# Patient Knowledge, Medication Adherence, and Influencing Factors: A Cross-Sectional Study among Hypertensive Patients in Greece

**DOI:** 10.3390/healthcare12090916

**Published:** 2024-04-29

**Authors:** Konstantinos Giakoumidakis, Evridiki Patelarou, Hero Brokalaki, Maria Bastaki, Nikolaos V. Fotos, Parthena Ifantopoulou, Antonios Christodoulakis, Anastasia A. Chatziefstratiou, Athina Patelarou

**Affiliations:** 1Department of Nursing, School of Health Sciences, Hellenic Mediterranean University, 71410 Heraklion, Greece; epatelarou@hmu.gr (E.P.); bastaki@hmu.gr (M.B.); achristodoul@hmu.gr (A.C.); apatelarou@hmu.gr (A.P.); 2Department of Nursing, School of Health Sciences, National and Kapodistrian University of Athens, 11527 Athens, Greece; heropan@nurs.uoa.gr (H.B.); nikfotos@nurs.uoa.gr (N.V.F.); a.chatziefstratiou@yahoo.gr (A.A.C.); 3417 NIMTS Veterans’ Fund Military Hospital, 11521 Athens, Greece; nopiifantopoulou@yahoo.gr

**Keywords:** patient knowledge, medication adherence, hypertensive patients

## Abstract

This study aimed to investigate the knowledge of patients with hypertension about their condition, adherence to antihypertensive medication, and the factors influencing it. A cross-sectional study was conducted in two cardiology outpatient clinics of two tertiary hospitals, in Greece. The study included 188 patients diagnosed with hypertension. The patients’ knowledge about their disease and adherence to medication were assessed by using the HK-LS and A-14 scales, respectively. Patients had sufficient knowledge levels about their disease, but significantly low levels of adherence to medication. Patients with higher knowledge levels were more adherent to medications [r(188) = 0.885, *p* < 0.001]. By using multivariate analysis, higher age (*p* = 0.018), residence in a more populous area (*p* = 0.041), more years with the disease (*p* = 0.012), and a lower number of medications (*p* = 0.03) were associated with higher levels of knowledge. Conversely, younger age (*p* = 0.036), lower educational levels (*p* = 0.048), fewer years with the disease (*p* = 0.001), and a higher number of medications (*p* = 0.003) were associated with lower adherence to medication. The Greek patients’ hypertension knowledge was sufficient; however, adherence to medication was significantly low. Healthcare managers could utilize our findings to design targeted interventions for improving adherence to medication for these patients.

## 1. Introduction

Arterial hypertension is a significant worldwide public health issue that affects over one-third of adults and is increasing in prevalence, especially in low and middle-income countries [1]. From 1990 to 2019, hypertension prevalence has increased twofold, rising from 650 million to 1.3 billion individuals [1]. Defined as having a blood pressure of 140/90 mmHg or higher, this widespread condition can result in stroke, heart attack, heart failure, kidney damage, and various other health complications [1,2]. Moreover, there are multiple risk factors associated with arterial hypertension, including aging, lifestyle modifications, and obesity [2]. Conversely, individuals with arterial hypertension often face the challenge of not having a complete understanding of their condition. This is due to factors such as insufficient knowledge, the presence of other medical conditions, concerns about side effects, non-pharmacological treatment, and making lifestyle changes [3,4,5].

In light of the critical importance of knowledge in the management of hypertension, studies have shown that providing patients with education and empowerment results in improved adherence to medication, and thus more favorable treatment outcomes [6,7,8,9,10]. Adherence refers to the degree to which an individual’s actions align with the prescribed recommendations from a healthcare professional, encompassing medication usage, dietary adherence, and lifestyle modifications [11,12]. Lack of proper adherence to medication for hypertension has various significant effects on its management, increasing the risk of cardiovascular events, poor renal outcomes (such as end-stage kidney disease), and even mortality, whereas studies have shown that better medication adherence substantially reduces cardiovascular risks and events, including drug-related side effects [13,14]. Therefore, it is crucial for patients to have high levels of medication adherence. However, that is not always the case, since there are multiple factors that affect medication adherence, including social and economic circumstances, self-efficacy, doctor–patient relationships, psychological aspects, complications related to hypertension, and various individual and social determinants [15,16,17,18,19,20].

Europe exhibits pronounced regional disparities for hypertension, with studies even suggesting a greater frequency and elevated mortality rates associated with stroke compared to North America [21]. For example, the EMENO National Epidemiological Study from Greece has revealed a significant prevalence of hypertension, specifically 39.6%, particularly among men and the elderly [22]. The current state of knowledge regarding the management of hypertension in Greece and the regional disparity in Europe necessitates the immediate development of a more comprehensive and robust information base. This is important, since the implementation of such an approach could contribute to the improvement of the overall health outcomes and quality of life of patients with hypertension in Greece.

The objective of this study was to investigate the levels of knowledge and adherence among patients with hypertension and to determine their correlation with the patients’ sociodemographic and clinical characteristics. Additionally, the study aimed to explore the relationship between patients’ knowledge and their medication adherence. This study was initiated to enhance the current information base on this subject and build a strong basis for future treatments and initiatives aimed at addressing any potential gaps in knowledge and adherence levels among individuals with hypertension.

## 2. Material and Methods

### 2.1. Study Design and Participants

A cross-sectional study was conducted among hypertensive patients who attended the cardiology outpatients’ clinics of two general hospitals in Greece, one military hospital in Athens and one tertiary university hospital in Heraklion, Crete Island. Sociodemographic and clinical patients’ characteristics were the independent variables of the present study, whereas hypertension knowledge and pharmaceutical treatment adherence were the dependent variables. Additionally, hypertension knowledge levels operated as an independent variable when its association with treatment adherence was examined.

The inclusion criteria of the study were as follows: (a) age ≥ 18 years old; (b) patients with diagnosed hypertension for at least six months; (c) patients receiving prescribed antihypertensive medications; and (d) sufficient knowledge of the Greek language. Patients who did not meet the above predefined criteria, those who inadequately completed the provided questionnaires, and those who did not consent to participate in the study were excluded.

In total, 217 patients attended the cardiology outpatient clinics of the two hospitals during the period of data collection. On the strength of the above-mentioned criteria, 188 patients (a participation rate of 86.6%) comprised the final study sample. Hence, 29 patients were considered ineligible to participate in our study. Specifically, 3 patients had been diagnosed with arterial hypertension for less than six months, 7 had insufficient knowledge of the Greek language, 11 failed to properly complete the questionnaires, and 8 patients declined to participate in our study.

### 2.2. Methods and Instruments

Data collection was carried out in June 2023. To assess patients’ knowledge of hypertension, the Greek version of the “Hypertension Knowledge Level Scale” (HK-LS) was used. Permission to use the questionnaire was obtained from its creators. The initial HK-LS, developed by Ekroc et al. (2012) and translated into Greek by Chatziefstratiou et al. (2015), consists of 22 statements [23,24]. Patients, completing the questionnaire, indicate whether each statement is correct, incorrect, or unknown. The statements cover six categories: disease definition (statements 1, 2), medication adherence (statements 3, 4, 5, and 12), medication (statements 6 to 9), lifestyle and habits (statements 10, 11, 13, 16, and 17), diet (statements 14, 15), and complications (statements 18 to 22). Only correct answers are scored, with a range of 0–22. Higher scores indicate sufficient knowledge, whereas lower scores indicate a knowledge deficit. The Cronbach’s alpha coefficient for the entire questionnaire was calculated at 0.67.

Furthermore, the Greek version of the “A-14 scale” [25] was used to assess patients’ adherence to medication. Also, permission to use the questionnaire was obtained from its creators. This instrument consists of 14 questions, and responses are given on a Likert scale with five levels, ranging from “never” (0) to “very often” (4) [26]. The total score ranges from 0 to 56. Patients with scores between 50 and 56 are considered 90% adherent, while those with lower scores are classified as non-adherent. The questions explore the following four areas of compliance: (a) patients’ ability to remember to take their medications (question 13); (b) patients’ adaptation to medication for safety and effectiveness reasons (questions 1–4, 6, and 7); (c) patients’ lifestyle, including financial burden and duration of treatment (questions 5, 8–10, and 14); and (d) patients’ attitude towards their medication (questions 11 and 12). The Cronbach’s alpha coefficient for the entire instrument was 0.88.

Last but not least, the authors created a special questionnaire to gather data on the socio-demographic and clinical characteristics of the study participants.

### 2.3. Ethics

The ethics committees of both the military (3/6/21 March 2023) and the university (20645/11 May 2023 and 21487/17 May 2023) hospitals granted written permission. Precautions were taken to protect participants’ privacy and anonymity, as well as the confidentiality of their data. Participants provided written and signed informed consent. The collected data were used solely for this study, and all research stages followed the ethical standards of the Helsinki Declaration of 1975, revised in 2013.

### 2.4. Statistical Analysis

We performed the statistical analysis using SPSS version 26.0 (SPSS Inc., Chicago, IL, USA). Continuous variables were expressed as mean ± standard deviation, and categorical variables were expressed as numbers and percentages. To correlate two continuous variables, we used the Pearson coefficient. Multiple linear regression analysis was used to examine the adjusted associations between independent variables and the scales HΚ-LS and A-14.

## 3. Results

The main socio-demographic and disease-related characteristics are depicted in Table 1. More than half of the participants were male subjects (52.1%), without university education (78.2%), living with partners (83.5%), in urban locations (77.1%), having an active vocational status (94.7%), having a family history of cardiovascular disease (62.8%) and non-active smokers (67%).

Additionally, the mean [±Standard Deviation (SD)] participants’ age was 68.6 (±10.7) years old (Table 2). Also, the sample’s mean (±SD) monthly income, years living with the disease, and number of prescribed medications were €859.7 (±362.3), 9.7 (±8.5), and 1.6 (±1.3), respectively, as shown in Table 2. As can be seen in Table 2, the mean (±SD) scores of HK-LS and A-14 were 18.5 (±2.2) and 11.1 (±5.7), respectively. Moreover, the Cronbach’s alpha coefficients for the entire HK-LS and A-14 questionnaires were calculated as 0.67 and 0.88, respectively.

The Linear Bivariate Correlation between the scores of the HK-LS questionnaire and the A-14 questionnaire recorded a statistically strong positive relationship [r(188) = 0.885, *p* < 0.001], as underscored in Table 3.

By using multivariate analysis, it was found that the main independent parameters of higher knowledge of hypertension were age (*p* = 0.018), place of residence (*p* = 0.041), years with the disease (*p* = 0.012), and number of medications taken (*p* = 0.003). Specifically, Table 4 indicates that higher age, residence in a more populous area, more years with the disease, and a lower number of medications correspond to a significantly higher level of knowledge compared to younger age, living in rural regions, fewer years with the disease, and taking a greater number of medications, respectively.

Also, as demonstrated in Table 5, the main strong predictors of improved adherence to pharmacological therapy were increased age (*p* = 0.036), higher educational level (*p* = 0.048), more years with the disease (*p* = 0.001), and a lower number of medications taken (*p* = 0.003).

## 4. Discussion

The knowledge of patients with arterial hypertension regarding their condition plays a pivotal role in effectively managing this chronic illness and fostering self-care behaviors, including adherence to pharmacological therapy. Identifying the factors influencing patients’ knowledge levels, alongside those directly impacting adherence to treatment plans, provides clinicians and administrators with insights into measures and policies tailored to individuals at a high risk of low knowledge and adherence levels, thus addressing problematic self-care behaviors. Our study’s findings reveal that hypertensive patients generally exhibit satisfactory knowledge levels about their condition and its management, whereas their adherence to prescribed pharmacological therapy remains notably low.

Our study underscores that patients with higher knowledge levels regarding their condition exhibit greater adherence to hypertension treatment. Additionally, factors such as advanced age, urban residence, longer disease duration, and fewer prescribed medications correlate strongly with elevated knowledge levels among patients with hypertension. Conversely, younger age, lower educational attainment, shorter disease duration, and a higher number of prescribed medications are associated with lower adherence levels. All patients were categorized as entirely non-adherent, as indicated by a maximum A-15 score of only 36. It is worth mentioning that the scale used for scoring ranged from 0 to 56, with scores between 50 and 56 representing 90% adherence.

As previously said, the participants in our study demonstrated sufficient levels of knowledge, as indicated by the mean score of 18.5 ± 2.2 on the HK-LS scale, which has a range of 0–22. Higher scores on this scale indicate a higher degree of knowledge. Variability in patients’ knowledge about hypertension is evident across different studies. For instance, one study revealed inadequate knowledge among over half of the participants [27], while another reported that 60% of patients possessed good knowledge [28]. Similarly, adherence levels vary, with one study finding that over half of hypertensive patients were not fully adherent [29], and another indicating medium–low adherence among 80% of participants taking multiple antihypertensive drugs [30]. Moreover, research on true resistant arterial hypertension revealed that 25% of patients did not adhere to prescribed medications [31]. Notably, better medication adherence has been associated with improved behavioral activation [32], while another study found high medication adherence rates among 63% of respondents [33]. These findings underscore the importance of addressing medication adherence in hypertension management and highlight the need for further research into associated factors.

Another significant finding of our study was the direct association between hypertension knowledge and adherence to medication, which underscores the importance of empowering patients with adequate knowledge for effective long-term disease management and self-care. In line with our results, enhancing patients’ knowledge of their condition is pivotal in promoting medication adherence, a finding supported by numerous studies [6,34,35,36].

Factors such as patients’ age, disease duration, and medication regimen significantly influence knowledge levels and adherence to pharmacological therapy. Older patients with longer disease duration and fewer prescribed medications exhibit superior disease knowledge and treatment adherence, aligning with previous research indicating higher knowledge scores among older individuals [37]. In addition, studies [38,39] have found that older age is linked to higher levels of medication adherence in hypertensive patients. This improved adherence may lead to better blood pressure management and a reduced risk of complications such as end-stage renal disease, as shown by [40]. Meanwhile, Ref. [41] revealed that younger patients exhibit diminished levels of adherence. However, contradictory findings exist, such as [42], which suggests that older age is a risk factor for poor medication adherence, emphasizing the need to consider age-related factors in adherence strategies.

Our study also reveals a significant association between higher knowledge levels and urban residence, echoing previous observations of knowledge disparities between urban and rural populations. Limited access to healthcare and educational resources in rural areas may contribute to lower hypertension knowledge levels, impacting self-care practices and medication adherence [43]. Efforts to bridge this gap should prioritize continuous health education, particularly in rural communities, where access to healthcare and educational resources may be limited [44]. Moreover, educational attainment influences medication adherence, with higher levels of education linked to better adherence to treatment regimens [45,46]. A higher level of education facilitates better disease understanding, a prerequisite for effective disease management and adherence to both pharmacological and non-pharmacological treatments. Interestingly, [30] found higher adherence rates among patients without complete secondary education, contrary to our findings, suggesting the need for further exploration.

### Limitations

Despite its merits, our study has a few limitations related to its design. Firstly, we used a cross-sectional research approach, which means that we could not establish causal relationships between knowledge, sociodemographic characteristics, and medication adherence. However, cross-sectional studies do provide valuable insights into the relationships between different variables and can help in designing future prospective/longitudinal studies. Therefore, future research could consider employing a longitudinal approach that includes additional measures such as mortality rates and overall quality of life. Secondly, the findings of our study cannot be generalized due to the limited sample size, since we only included patients with hypertension from two tertiary hospitals in Greece. Thirdly, medication adherence was assessed using self-reported data, indicating a lack of objective measurements. Finally, in terms of reliability, it is worth noting that while the Cronbach’s alpha coefficient for the Health Knowledge Literacy Scale (HK-LS) showed commendable internal consistency, the same measure for the A-14 tool, although considered acceptable [47], may be open to interpretation and potentially questionable. As a result, caution is advised when interpreting adherence findings due to concerns about reliability. However, the low medication adherence rates we observed highlight the importance of addressing this issue in patient care. Future research could focus on refining measurement tools or exploring alternative methods to enhance adherence assessments in patients with arterial hypertension.

## 5. Conclusions

In conclusion, this study suggests that patient knowledge is a significant factor in effectively managing arterial hypertension and fostering adherence to pharmacological therapy. The findings further underscore the importance of empowering patients with a comprehensive understanding to facilitate long-term disease management and self-care practices. Factors such as age, urban residence, disease duration, and medication regimen significantly influence both knowledge levels and adherence behaviors, emphasizing the need for tailored interventions to address diverse patient demographics. It is important to note that older patients who have had the disease for a longer time and take fewer prescribed medications tend to have greater knowledge and adherence, which emphasizes the complex nature of managing hypertension.

Consequently, healthcare providers could play a pivotal role in implementing continuous health education initiatives, especially in rural areas, to bridge knowledge gaps and enhance medication adherence among hypertensive populations. By addressing knowledge gaps and adherence barriers, healthcare providers can significantly contribute to improving hypertension management outcomes and reducing associated complications. Finally, the study’s limitations suggest the need for further research into effective educational interventions and new adherence assessment tools that could enhance patient care.

## Figures and Tables

**Table 1 healthcare-12-00916-t001:** Socio-demographic and disease-related characteristics.

		Ν	N %
Gender	Men	98	52.1%
	Women	90	47.9%
Educational level	Up to Post-secondary education	147	78.2%
	Higher Education/Holder of MSc or PhD	41	21.8%
Type of living	Alone	31	16.5%
	Living with others	157	83.5%
Place of residence	Civil center	145	77.1%
	Countryside	43	22.9%
Working status	Worker	178	94.7%
	Unemployed	10	5.3%
Family history of cardiovascular disease	Yes	118	62.8%
	No	70	37.2%
Smoking	Smoker	62	33.0%
	Non-smoker or former smoker	126	67.0%

**Table 2 healthcare-12-00916-t002:** Other characteristics.

	Mean	Standard Deviation	Range
Age (years)	68.6	10.7	51
Monthly income (euro)	859.7	362.3	3000
Years with the disease	9.7	8.5	48
Number of medicines	1.6	1.3	9
HΚ-LS Score	18.5	2.2	11
A-14 Score	11.1	5.7	36

**Table 3 healthcare-12-00916-t003:** Linear Bivariate Correlation between the total scores of HK-LS and A-14 questionnaires.

		HΚ-LS Score	A-14 Score
HΚ-LS Score	Pearson Correlation	1	0.885 **
	Sig. (2-tailed)		0.000
	N	188	188
A-14 Score	Pearson Correlation	0.885 **	1
	Sig. (2-tailed)	0.000	
	N	188	188

** Correlation is significant at the 0.01 level (2-tailed).

**Table 4 healthcare-12-00916-t004:** Multivariate analysis between the demographic characteristics and HΚ-LS score of the participants.

Model	Unstandardized Coefficients ^a^		Standardized Coefficients ^a^	t	Sig.	95.0% Confidence Interval for B	
	B	Std. Error	Beta			Lower Bound	Upper Bound
(Constant)	16.991	2.040		8.329	0.000	12.965	21.017
Gender	−0.411	0.343	−0.095	−1.199	0.232	−1.089	0.266
Educational level	0.676	0.403	0.130	1.677	0.095	−0.120	1.472
Type of living	0.140	0.455	0.024	0.308	0.758	−0.758	1.039
Place of residence	−0.789	0.384	−0.154	−2.058	**0** **.041**	−1.546	−0.032
Working status	0.466	0.694	0.049	0.672	0.502	−0.903	1.835
Family history of cardiovascular disease	−0.261	0.327	−0.059	−0.800	0.425	−0.906	0.383
Smoking	−0.502	0.355	−0.109	−1.412	0.160	−1.203	0.200
Age	0.039	0.016	0.193	2.379	**0.018**	0.007	0.071
Monthly income	0.000	0.000	0.025	0.333	0.740	−0.001	0.001
Years with disease	0.052	0.020	0.202	2.549	**0.012**	0.012	0.092
Number of medicines	−0.385	0.128	−0.224	−3.014	**0.003**	−0.636	−0.133

^a^. Dependent variable: HΚ-LS score. Bold indicates the statistically significant values at a level of 5%.

**Table 5 healthcare-12-00916-t005:** Multivariate analysis between the demographic characteristics and A-14 score of the participants.

Model	Unstandardized Coefficients ^a^		Standardized Coefficients ^a^	t	Sig.	95.0% Confidence Interval for B	
	B	Std. Error	Beta			Lower Bound	Upper Bound
(Constant)	6.060	5.316		1.140	0.256	−4.431	16.550
Gender	−0.852	0.894	−0.075	−0.953	0.342	−2.617	0.913
Educational level	2.089	1.051	0.152	1.987	**0** **.048**	0.015	4.164
Type of living	−0.247	1.187	−0.016	−0.208	0.835	−2.589	2.095
Place of residence	−0.311	1.000	−0.023	−0.311	0.756	−2.283	1.662
Working status	1.421	1.808	0.056	0.786	0.433	−2.146	4.989
Family history of cardiovascular disease	−1.234	0.851	−0.105	−1.450	0.149	−2.913	0.446
Smoking	−1.496	0.926	−0.124	−1.615	0.108	−3.323	0.332
Age	0.090	0.043	0.170	2.110	**0** **.036**	0.006	0.174
Monthly income	0.001	0.001	0.070	0.935	0.351	−0.001	0.003
Years with disease	0.181	0.053	0.269	3.421	**0** **.001**	0.076	0.285
Number of medicines	−1.014	0.333	−0.224	−3.049	**0** **.003**	−1.670	−0.358

^a^. Dependent variable: A-14 score. Bold indicates the statistically significant values at a level of 5%.

## Data Availability

The data that support the findings of this study are available from the corresponding author upon reasonable request.

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
