# Peer review of "Patient Knowledge, Medication Adherence, and Influencing Factors: A Cross-Sectional Study among Hypertensive Patients in Greece"

_healthcare, 2024, doi:10.3390/healthcare12090916_

Round 1

Reviewer 1 Report

Comments and Suggestions for Authors

I enjoyed reading this manuscript. This study provides important information for creating further programming to improve hypertension in your community. 

Line 55-58: The two sentences related to non-adherence to medications seem to be explaining the same concept. Please condense these two sentences into one. Consider adding medication non-adherence can increase risk of adverse drug events.

Line 61: “self-confidence”: would self efficacy be a better term?

Line: 73: delete “therefore,”

Line 83: was the study IRB approved? Was informed consent collected?

Line 90: state the time period of the study

Line 97-98: I do not understand this sentence. If this is a strength, perhaps mention it in discussion section. Does it mean 188 patients enrolled into the study? Please clarify.

Line 133: Please add # participants recruited, # participants consented, # participants completed the surveys, # incomplete surveys.

Line 134: is 52.1% “most”? I’d say, “more than half”

Line 149: can you also provide the range in Table 2?

Line 158-161: Please clarify what is this compared to?

Line 162 Table 4: can you define # of medicines? Does it include only antihypertensives or all medications? Can you define place of residence, how do you categorize them?

Line 183: the definition of non-adherence should be earlier in the method section

Line 168-231: How about cost of medication? And smoking history? Have you notice any difference in income and smoking history in urban vs rural?

Line 233-249:

Additional limitations to consider:

1. Lack of objective measurements in medication adherence. The instruments are self-reported by the patient. Using two methods (pharmacy fill records or pill box count) and patient surveys will be more robust.

2. Are there patients who are illiterate and cannot complete the survey?

Comments on the Quality of English Language

Excellent, easy to read and understand. 

Author Response

Dear reviewer,

I am writing to thank you for your fruitful comments that contribute to the improvement of the submitted manuscript. Please find attached a step-by-step response to your valuable comments.

Once again, many thanks.

Best regards,

Prof. Konstantinos Giakoumidakis

Reviewer 2 Report

Comments and Suggestions for Authors

In this study, the authors provide results of a cross-sectional study aiming at investigating the knowledge of patients with hypertension about their clinical condition, factors influencing it, and adherence to treatment. For the study, the authors used a well set of inclusion and exclusion criteria , a version of the Hypertension Knowledge Level Scale revised in agreement with the creators of the HK-LS to assess the patients' understanding and knowledge of their medical condition, and 

.  The Hypertensive patients recruited for the study were pooled from the cardiology outpatients' clinics of two general hospitals in Greece, 1 military hospital in Athens, and one tertiary university hospital in Crete island. The conclusion of the authors, supported by the data reported in the manuscript indicate that overall patients had sufficient knowledge about their disease but significant low level of medication compliance. Patients with higher knowledge levels presented with better compliance. Additionally, the less medications the patients were using the better was the compliance, and the knowledge levels were definitely better in urban settings that in rural settings. How these findings will be used moving forward to design more targeted interventions was beyond the scope of this study and needs to be addressed in future studies. 

Comments

The study provide important results that will hopefully be properly utilized in Greece to better target hypertensive patients with more effective therapeutic approaches. The study is properly written.

My main comments is that the authors should clearly provide the number of patients used for the study and what was the attrition rate of the study. It would appear that they recruited 188 patients (based on Table 1, male and female distribution), but it is not clear whether this number is already the result of any attrition rate. In other words, how many patients were recruited? How many responded to the questionnaires, and how many patients continued to report and provide information used in the study? This information would provide a better sense of how big future studies need to be and how long those studies can be carried out longitudinally in order to obtain valid data considering the potential attrition rate.

Author Response

Dear reviewer,

I am writing to thank you for your fruitful comments that contribute to the improvement of the submitted manuscript. Please, find attached a step-by-step response to your valuable comments.

Once again, many thanks.

Best regards,

Prof. Konstantinos Giakoumidakis
